# Investigating Non-Pharmacological Stress Reduction Interventions in Pediatric Patients Confirmed with Salivary Cortisol Levels: A Systematic Review

**Maria Grigoropoulou [1], Emmanouil I. Kapetanakis [2,\*], Achilleas Attilakos [3], Anestis Charalampopoulos [4], Anastasia Dimopoulou [5], Efstratios Vamvakas [6], Eleftheria Mavrigiannaki [5] and Nikolaos Zavras [5]**

1   Second Health Centre of Peristeri, 12131 Athens, Greece; maria_grigoropoulou@yahoo.com
2   Department of Thoracic Surgery, National and Kapodistrian University of Athens, "Attikon" University Hospital, 12462 Athens, Greece
3   Third Department of Pediatrics, National and Kapodistrian University of Athens, "Attikon" University Hospital, 12462 Athens, Greece; aattil@med.uoa.gr
4   Third Department of Surgery, National and Kapodistrian University of Athens, "Attikon" University Hospital, 12462 Athens, Greece; achalaral@med.uoa.gr
5   Department of Pediatric Surgery, National and Kapodistrian University of Athens, "Attikon" University Hospital, 12462 Athens, Greece; natasa_dimo@hotmail.com (A.D.); ele17mav@yahoo.gr (E.M.); nzavras@med.uoa.gr (N.Z.)
6   Second Department of Critical Care, National and Kapodistrian University of Athens, "Attikon" University Hospital, 12462 Athens, Greece; evamvakas@yahoo.gr
*   Correspondence: emmanouil.kapetanakis@gmail.com; Tel.: +30-210-5831195

**Abstract:** For many children, hospitalization can lead to a state of increased anxiety. Being away from home, the invasive procedures undertaken, and the uncertainty of the outcome cause an uncomfortable situation in anticipation of real or imagined hazards. This systematic review aims to assess current evidence on the types of non-pharmacological interventions used and their impact on children's anxiety or distress levels when they visit the hospital for planned or unplanned admissions. The Databases PubMed, Psych INFO, and Google Scholar were queried for papers published from January 2000 to March 2023 reporting the use of non-pharmacological interventions interacting with children in hospital or clinical environments and confirmed with saliva cortisol levels. A total of nine studies were retrieved. Across these studies, four different strategies of non-pharmacological interventions were used. Anxiety and distress were found to be reduced in the majority of the studies as confirmed with salivary cortisol. Overall, there is evidence that non-pharmacological interventions hold a promising role in reducing levels of anxiety or distress in children as confirmed with saliva cortisol. However, research on saliva cortisol as a tool of anxiety measurement requires higher quality studies to strengthen the evidence base.

**Keywords:** anxiety; hospital; non-pharmacological interventions; saliva cortisol; children

## 1. Introduction

Hospitalization and surgery are significant events in the lives of children [1,2]. Separation from their family environment combined with medical procedures and the absence of control of the situation creates fear and anxiety [3,4]. Any painful procedures such as needle exposure, operations, destruction of body image or immobility have been established worldwide as negative experiences by children [4].

Predictors of increased anxiety include negative memories of previous hospital experiences or pediatrician visits and can last into adolescence, affecting biologically based vulnerabilities, temperament, attachment style, and quality of parent-child relationships. Children of anxious parents who use avoidance coping mechanisms and of divorced or

separated parents are more anxious. Other factors also include gender of the parent (mothers are more anxious), repeated hospitalizations, the baseline temperament of the child, and also perioperative environmental factors. [5].

Furthermore, excessive anxiety also impedes a child's efficacy to deal with medical treatments, increases their aversion to pain, and promotes an uncooperative behavior and negative emotions towards healthcare professionals [6].

The Hypothalamic-Pituitary-Adrenal axis (HPA Axis) is an essential part of the neuroendocrine system that controls stress reactions [7]. Cortisol is the most active glucocorticoid of the neuroendocrine system [8] distributed in all body fluids such as blood, urine, and saliva after stimulation of the hypothalamic cells [9]. The release of cortisol under normal conditions follows a circadian rhythm, this means it is characterized by an early morning peak "steepest in the first three hours after awakening" followed by a gradual decline throughout the day "reaching its lowest values around midnight" [10]. In physiological conditions, the rhythm of release of cortisol is stable since is not affected by age, gender or body composition [11,12]. However, stressful situations such as fear and anxiety increases the activity of the HPA Axis which sequentially increases the release of cortisol [13]. Studies have shown that salivary cortisol (SC) levels correlate strongly with the unbound cortisol in the blood and consequently it may be used as a reliable tool to determine the effectiveness of interventions utilized to reduce stress in different medical conditions [14].

Various types of non-pharmacological interventions have been used in order to reduce anxiety in hospitalized children [15,16]. These strategies could be categorized as educational, behavioral, parental presence, and complimentary/alternative techniques, they are not invasive, they carry low risk of adverse events, and are inexpensive. Recently, in the array of non-pharmacological strategies, multimedia apps and games on the internet and on mobile electronic platforms have been added, providing additional psychological support to hospitalized children and their parents [15].

This systematic review aims to investigate the non-pharmacological interventions used to minimize anxiety in hospitalized children, and confirmed with SC levels.

## 2. Materials and Methods

### 2.1. Study Design

This review was conducted according to the Preferred Reporting Items for Systematic Reviews and Meta-Analyses (PRISMA) guidelines [17] and methodological recommendations from the Cochrane Handbook for Systematic Reviews and Interventions [18]. The protocol for this systematic review was registered on the National Institute for Health and Care Research PROSPERO international prospective register of systematic reviews (CRD42023409263) and can be accessed at https://www.crd.york.ac.uk/prospero/display_record.php?ID=CRD42023409263 (accessed on 5 April 2023).

### 2.2. Search Strategy

Relevant hypothesis articles were searched by using the databases of PubMed, Google Scholar, ScienceDirect and APA PsycInfo®. Gray literature was also searched for additional articles. This review covers a period between 2000 to 2023 in order to elucidate the most recent developments in the related issue. The Mesh terms used in the search strategy were: [perioperative anxiety] AND [non-pharmacological interventions] AND [perioperative anxiety OR stress OR fear] AND [hospitalization OR surgery] AND [saliva cortisol] AND [children]. The search was conducted in accordance with the PICOS (Population, Intervention, Comparison, Outcome, Study) [19] criteria as follows:

- P: the population was children aged 3–18 years, who were visiting the hospital with any physical health condition. There were no limitations on their gender or socioeconomic characteristics.
- Intervention: any interactive non-pharmacological strategy utilized.

- Comparison: the comparison was between CS levels of patients undergoing the intervention (Intervention Group, IG) and control group (CG).
- Outcome: saliva cortisol levels before and after intervention in the IG and CG.
- Study Design: randomized, quasi-randomized controlled trials and cohort studies.

### 2.3. Inclusion and Exclusion Criteria

The inclusion criteria assessed for this systematic review were as follows: (1) randomized or quasi-randomized trials conducted to assess non-pharmacological strategies to lower perioperative anxiety and stress in pediatric patients and confirmed by SC levels, (2) publications in English language, (3) availability of the full texts, and (4) children aged $\geq 3$ years and $\leq 18$ years. The age range of 3 to 18 years was selected because progress in children aged >3 years is more clearly visible in many aspects of executive function such as inhibitory control, working memory, cognitive flexibility, and simple planning [20]. Exclusion criteria were as follows: (1) article types such as abstracts, reviews, editorial letters, case reports, case series, and meta-analyses, (2) studies not published in the English language, (3) children with attention deficit hyperactivity disorder, autism, and Down syndrome because there are many additional psychological and physiological issues to take into account when preparing these children. In addition, children needing dental care were excluded from the analysis because conventional dental care does not require extensive surgical interventions contrary to surgical procedures which are more complicated and elicit a higher anxiety level. For example, in a study by Dereci et al., the Dental Fear Scale (DFS) score was significantly increased in the third molar group compared to the dental extraction group ($p < 0.05$) [21]. Even simple dental extraction illicit lower anxiety as the study by Kumari et al. demonstrated by comparing preoperative salivary cortisol levels three days before major surgery under general anesthesia (study group) undergoing surgery for trauma, orthognathic surgery, and other pathologies such as marsupialization of a larger cyst or removal of tumor and to compare it with the patient's undergoing extraction under local anesthesia (control group). The results demonstrated that the comparison between salivary cortisol in the study group before ($18.9 \pm 23.7$) and after ($23.7 \pm 9.2$) surgery were significantly higher than the control group ($15.2 \pm 5.5$) ($p = 0.013$ and $p = 0.005$), respectively [22]. (4) Treatment strategies based on pharmacological or combined pharmacological and non-pharmacological therapies, and (5) administration of oral or intravenous steroids for various diseases. Duplicate studies were assessed by EndNote and were removed manually. The identification and selection process is shown in Figure 1.

### 2.4. Study Selection

Two of the authors (MG and NZ) separately screened the titles and abstracts of the selected articles initially followed by the extraction of the full text article when a given abstract was considered of potential interest. Any disagreement was resolved by amicable discussion between the two authors.

### 2.5. Data Extracted

Author(s), year and journal of publication, country, number of patients, age (median, mean, range), type of intervention, and outcomes were extracted from all selected articles. The collected data were recorded in a Microsoft Excel spreadsheet database (Microsoft Corporation, 2016) secured in a password protected computer.

### 2.6. Classification of Non-Pharmacological Interventions and Definitions

The non-pharmacological interventions recorded were classified accordingly as:

1. Educational intervention was defined as any approach which provided information, explanation, and preparations relevant to any surgical or medical intervention.
2. Behavioral intervention was defined as the strategy which encouraged children to relieve pain or anxiety from a painful event.

3.  Parental presence was a strategy to reduce a child's anxiety by parental comforting and attendance during procedures such as induction of anesthesia.

4.  Complementary and alternative strategies were defined as interactive approaches constructed to be performed alone or with other techniques to relieve anxiety of fear feelings during any medical or perioperative procedures [15].

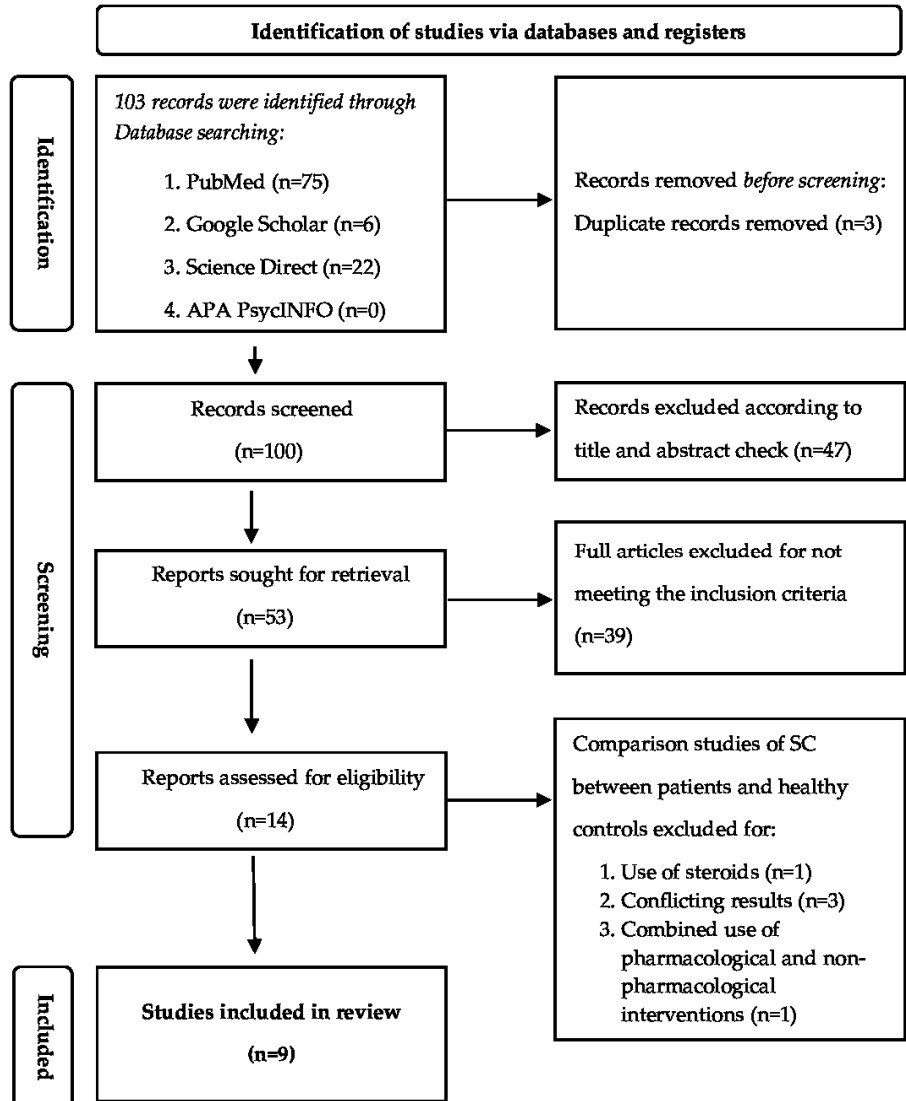

**Figure 1.** PRISMA flowchart of the article identification and selection steps.

### 3. Results

Of the 103 records identified, 100 were considered after duplicates were removed. The 1st selection phase (titles and abstracts screening) resulted in 16 articles for full text evaluation. The 2nd selection phase resulted in nine articles [23–31], eligible for the final qualitative analysis.

### 3.1. Studies Characteristics

The sample size of the studies ranged from 29 to 184 patients. Of the nine studies, three studies were conducted in Brazil [25,26,31], two in Turkey [28,29], and one in Sweden [23], Italy [24], USA [27], and Peru [30] (Table 1).

**Table 1.** Patients' characteristics and interventional procedures.

| Authors/Year | Age: Range or Mean ± SD Years or Months | N [Gender (M/F)] | Reasons of Hospitalization | Type of Intervention | Outcome Results Based on SC Levels |
|---|---|---|---|---|---|
| Wennström et al. [23], 2011 | IG: 94 ± 23 (months) CG: 91 ± 26 (months) | 93 [79/14] | Day-surgery operations | IG: PD CG1: standard care CG2: standard care + information | Significantly lower SC levels (*p* = 0.003) an in the PD group post-op |
| Calcaterra et al. [24], 2015 | IG: 8.59 ± 3.7 (yrs) CG: 7.36 ± 2.48 (yrs) Range: 7–17 (yrs) | 40 [32/8] | Day-surgery operations | IG: AAT CG: common care | No statistically differences between groups (*p* = 0.70) |
| Lima et al. [25], 2015 | Range: 6–14 (yrs) | 40 [NI] | Urological surgery | IG: Interactive Musical Activities CG: no music | No differences in SC levels between groups (*p* > 0.05) |
| Saliba et al. [26], 2016 | 6–7 (yrs) | 36 [NI] | Appendicitis, femur fracture, adenoid hypertrophy, trachea trauma, phimosis, paraphimosis, pneumonia | IG: CD at lunch CG: CD at dinner | SC levels were significantly reduced after CD intervention in both groups (*p* < 0.01 and *p* < 0.01, respectively Better satisfaction for CD at lunch |
| Branson et al. [27], 2017 | IG: 13.43 ± 0.59 CG: 12.83 ± 0.58 *p = 0.364* | 48 [24/24] | Surgical, pneumological, trauma, neurological, immunological, endocrine, psychiatric, gastrointestinal, and liver disorders | IG: AAA CG: no person or live dog | Increased positive effect and decreased negative effect in the AAA group compared to CG, but not significant. No statistical differences in SC levels between groups (*p* = 0.47) |
| Ozdogan et al. [28], 2017 | IG: 7.41 ± 2.30 (yrs) CG: 7.37 ± 2.06 (yrs) *p = 0.94* | 48 [21/27] | Tonsillectomy and or adenoidectomy | IG: Mother's presence CG: Mother's absence | SC levels were significantly increased in CG after induction and in recovery room compared to IG (*p* = 0.001 and *p* = 0.02, respectively) |
| Volkan et al. [29], 2019 | IG: 11.55 ± 2.52 CG:11.44 ± 2.66 *p = 0.773* | 184 [85/99] | Gastroscopy | IG: Detailed information CG: brief information | SC levels significantly less in the IG (*p* < 0.001) |
| Alarcon-Yacketto et al. [30], 2021 | IG: 9.35 ± 1.59 CG: 10.33 ± 1.39 p: 0.091 | 29 [18/11] | Various types of acute appendicitis, bone fractures, mastoiditis | * IG: Augmented reality books # CG: standard books | Marked decreased in SC levels in IG vs. CG (*p* < 0.001) |
| Brockingston et al. [31], 2021 | Total: 7.06 ± 2.23 (yrs) IG: 7.02 ± 2.14 (yrs) CG: 7.1 ± 2.34 (yrs) | 81 [40/41] | PICU (respiratory diseases most common) | IG: Storytelling CG: Riddle | Marked decreased of SC in the IG vs. CG (*p* < 0.001) |

N: Total sample size, * IG: Interventional group, # CG: Control group, AAT: animal-assisted therapy, AAA: animal-assisted activities, PD: perioperative dialogue, CD: clown doctors.

### 3.2. Study Population

A total of 599 children were included in this review. From the available data [23,24,27–31] a slight preponderance of boys over girls was observed (299 boys vs. 224 girls).

### 3.3. Types of Intervention

3.3.1. Educational Intervention

Two studies [23,29] used an educational approach by providing perioperative information. Wennström et al. [23] evaluated the perioperative dialogue (PD) versus standard care and standard care plus perioperative information. The intervention was performed

in three steps: preoperatively, intraoperatively and postoperatively by the same nurse. Preoperatively, the nurse listened to the thoughts and questions of the child and helped the child to gain a sense of control in facing the unknown situation. Intraoperatively, the dialogue was continued further, while postoperatively the nurse evaluated the child's experience. In the control groups no PD was performed. A repeated measurement design was used to monitor intra- and inter-individual differences of cortisol concentration in saliva. Cotton-based neutral Salivette tubes (Sarstedt$^{TM}$, Landskrona, Sweden) were utilized. A swab was chewed and then placed in a sterile plastic tube. The Salivette tubes were then centrifuged at $1711 \times g$ for 15 min at 20 °C and then frozen at minus 80 °C until assayed simultaneously. A commercial radioimmunoassay based technique for salivary cortisol was used (Spectria$^{TM}$, Cortisol I125, Landskrona, Sweden) for measuring cortisol level.

In a study by Volkan et al. [29], preparation of the IG included a brief explanation about "what is endoscopy", "why is required", and the steps of the endoscopic procedure. Furthermore, a detailed explanation of the procedure was performed by the pediatric gastroenterologist and the endoscopy nurse. In the CG, only general information was given. To determine the basal cortisol levels (bSCL), saliva specimens were collected at home in the morning hours (08.00–12.00 AM) the day before the endoscopy and brought by the parents to the hospital. On the day of the endoscopy, the patients arrived at the hospital during the morning hours and the procedure was performed until 12.00 AM. The s-cortisol levels (in ng/mL) were determined using an enzyme-linked immunosorbent assay (ELISA) method [29].

### 3.3.2. Behavioral Interventions

In two studies [30,31] the distraction strategy was used. The first study assessed the consequences of reading augmented reality (AR) books [30] on SC levels in hospitalized children for various surgical conditions (Table 1). In IG patients, a book AR was given, while in the CG a standard book without a tablet was offered. SC sampling was taken before and after each intervention. In addition, a visual analogue scale (VAS) was used for assessing the psychological stress of each group [30].

In the second study [31], the authors randomized patients hospitalized in ICU for similar medical conditions into two groups. In the IG (storytelling group) there were given selected stories to the children. At any time, children could change the story or request another story. In the CG, a riddle question (riddle group) was given. The duration of session in each group lasted approximately, 25–30 min. SC samples were taken one minute before the intervention and immediately after the intervention.

### 3.3.3. Maternal Presence

In this study [28] the patients of the CG were separated from their mothers at the preoperative evaluation room, while the mothers of the IG stayed with their children till the induction of anesthesia. SC samples from both groups were evaluated in four different phases: (1) in preparation room, (2) after anesthesia induction, (3) at the 30th min of operation, and (4) in the recovery room.

### 3.3.4. Complementary Interventions

Four studies [24–27] assessed the impact of complementary interventions on perioperative stress in hospitalized children.

Animal-Assisted Therapy

a.   In the study by Calcaterra et al. [24], the authors evaluated the impact of animal-assisted therapy (AAT) on the children's response to stress and pain in the immediate postoperative period. They randomly assigned 40 patients into two groups: IG (20 patients) and CG (20 patients). The primary end-point concerned the impact of AAT in IG on neurological signs as measured with electroencephalogram. Secondary end-points concerned the impact of AAT on the cardiovascular system by measuring vital signs (heart rate, blood pressure, oxygen saturation, and cerebral prefrontal

oxygenation) and endocrinological impact as measured by SC levels at two hours after surgery (T1), at 20 min following T (T2), and only for SC levels at midnight (T3), the time when cortisol is normally at its lowest. Additionally, a faces pain scale was used to measure child self-reported pain at T1 and T2.

b.  Branson et al. [27] assessed the benefits of AAT on biobehavioral stress responses (anxiety, positive, and negative affects) and SC levels in hospitalized children (Table 1) in the IG versus CG, before and after intervention.

Music Therapy

Lima et al. [25] investigated the effect of interactive musical activities in reducing anxiety in children hospitalized for urogenital anomalies. The IG participated in 15–30 min daily sessions for about 5 days of hospitalization, except on the day of surgery. Two SC samples at 8.00 AM and 4 PM, from the first day of hospitalization until at least the first postoperative day were taken. The CG did not participate in musical activities sessions.

Clown Doctors (CD)

Saliba et al. [26] correlated entertainment CD activities on hospitalized children for various pathological and surgical conditions with SC. The authors took SC samples and measured VAS prior CD activities, followed by another SC sample and VAS after CD activities. The CD performed their activities at lunch (lunch CD) and dinner (dinner CD).

### 3.4. Effect of Non-Pharmacological Interventions on SC Levels in Hospitalized Children and Adolescents

Six [23,26,28–31] of the nine studies (66.6%) showed that non-pharmacological interventions reduced significantly SC levels.

#### 3.4.1. Educational Intervention

Wennström et al. [23] reported a statistically significant reduction in SC levels before and after intervention among patients who underwent PD versus those with standard care with preoperative information ($p = 0.006$) or without preoperative information ($p = 0.003$). Volkan et al. [29] demonstrated that the preparatory information in children and adolescents resulted in significantly lower SC levels pre- and post-endoscopy procedures between IG and CG ($p < 0.001$ and $p = 0.001$, respectively)

#### 3.4.2. Behavioral Intervention

In terms of distraction techniques either with AR books [30] or with storytelling [31], Alarcón-Yaquetto [30] reported a significant decreased of SC levels after intervention with AR books ($p = 0.019$). Similarly, Brockington et al. [31] showed a marked decrease in SC levels in the storytelling group ($p < 0.001$).

#### 3.4.3. Maternal Presence

In the study by Ozdogan et al. [28] the presence of the mother during anesthesia resulted in significant lower SC levels after induction and in the recovery room ($p = 0.001$ and $p = 0.02$, respectively).

#### 3.4.4. Complementary Interventions

Calcaterra et al. [24] and Branson et al. [27], who examined the impact of animal-assisted interventions in children which underwent either daily surgical operations [24] or suffered from surgical and/or pathological conditions [27], failed to show statistical differences in SC levels after sessions with pet interventions ($p = 0.7$ and $p = 0.47$). Similarly, Lima et al. [25] reported no significant differences in SC levels of children participated daily in interactive musical activities during hospitalization when compared to CG ($p > 0.05$). Contrary, by using a program with CD, Saliba et al. [26] reported significant differences in SC levels at different sessions (at lunch and dinner, respectively, $p < 0.01$ and $p < 0.01$, respectively).

### 3.4.5. Additional Findings

In the study by Wennström et al. [23], the beneficial influence of PD was confirmed further by lower consumption of morphine postoperatively in the IG compared to the CG ($p = 0.014$). Calcaterra et al. [24] observed faster neurological activity ($p < 0.001$) and lower pain perception ($p = 0.01$) in the postoperative period in children undergoing AAT intervention compared to CG. Lima et al. [25] did not find significant differences regarding positive and negative feelings related to the surgery. Saliba et al. [26] reported a better satisfaction after introducing intervention with CD at least in one measurement ($p < 0.01$). Although increased positive changes related to negative in the IG were noted, Branson et al. [27] did not find statistically significant differences regarding pre- vs. post-changes between IG and CG in the following parameters: positive effects ($p = 0.76$), negative effects ($p = 0.35$), state anxiety ($p = 016$), and CRP ($p = 0.18$). Ozdogan et al. [28] reported no statistically significant difference between groups regarding maternal education levels ($p = 0.45$). In the study by Volcan et al. [29], after preparatory information, the IG experienced less anxiety scores, less duration of sedation and endoscopy, lower propofol doses and reduced duration of recovery ($p < 0.001$, $p = 0.003$, $p = 0.016$, $p < 0.001$ and $p < 0.001$, respectively). Alarcon-Yacketto et al. [30] reported a statistically significant increase in VAS score, i.e., less emotional stress after the AR intervention ($p < 0.001$). Finally, Brockington et al. [31] reported statistical significant differences regarding oxytocin and pain scores between IG and CG ($p < 0.001$ and $p < 0.01$, respectively).

## 4. Discussion

This systematic review demonstrated that six out of nine (66.6%) non-pharmacological interventions are effective in reducing anxiety, as confirmed by SC samples, in children which were hospitalized for various medical and surgical conditions.

Preoperative preparation based on information is a process that gives the opportunity to inform the patient about the procedure of surgical intervention. Although the mechanism of the effect of preoperative information is not well understood, it seems that is an important tool for the effective management of children undergoing medical procedures [32]. The usual practice involves a visit to the operating room, a demonstration of the surgical instruments, a description or visual depiction of the procedure [15]. However, the implementation of more sophisticated practices with the use of modern technology have shown significantly lower anxiety scores and disturbances during the perioperative period. For example, Hatipoglu et al. [33] investigated the effect of audiovisual and auditory presentations on preoperative anxiety and behavioral disturbances. They found less m-YPAS anxiety scores in children received audiovisual presentation on preoperative preparation compared to children with auditory presentation and control group informed with the usual anesthetic practice ($p < 0.001$ and $p < 0.001$, respectively). These results are further supported by the study by Tomazsek et al. [34] who found that anxiety measured with the State-Trait Anxiety Inventory for Children (STAI-CH; range: 20–60 pts) or the State-Trait Anxiety Inventory for Adolescents (STAI; range: 20–80 pts) in children receiving detailed information was significantly lower than the control group ($p < 0.001$). Our findings [23,29] are in line with current evidence that preoperative preparation in of importance in minimizing the postoperative stress in children.

Distraction is a cognitive behavior therapeutic technique that may be used in children to refocus the attention away from pain and anxiety [15,35]. Distraction as a coping strategy use cognitive and behavioral aspects. It has been shown as the best strategy to divert the attention of children from pain and anxiety [35]. Distraction can be divided into two types: passive and active. A paradigm of passive distraction is when the health care professional calls the child to remain quiet while the health care professional is actively distracting the child (i.e., by talking, reading a book or singing). Active distraction, on the other hand, encourages the child's participation in the activities during the procedures [16]. Although active distractions seem to be superior than passive distractions because they demand multisensory engagement that interrupts multisensory pain stimuli, the comparison of

these methods has led to inconclusive results [35]. For example, MacLaren and Kohen [36] reported that active forms of distractions may be too exhausting for some children experiencing pain, whereas a passive technique may be more effective. Our review is in line with two studies showing [30,31] reduction in anxiety after AB-books and storytelling intervention's, confirmed with SC levels hospitalized children. The impact of parental present as a strategy carries conflicting results. Kain et al., suggest that feared and anxious children with calm parents may benefit from the parental presence of one or both parents [37]. In addition, Yoo et al., by investigating the preoperative visual information, and maternal presence during induction in anesthesia, found less anxiety during recovery from anesthesia, when compared to the control [38]. Similarly, Ozdogan et al. [28] reported less postoperative stress as determined by the SC levels. Contrary, a number of other studies [39–42] did not demonstrate beneficial results in terms of anxiolysis or postoperative outcomes. It is noteworthy that Zand et al. [43] reported that the presence of a parent during induction in anesthesia had the same results as midazolam in decreasing the incidence of postoperative agitation.

Complementary interventions and alternative medicine involve strategies that are not contemplated as conventional or well established. They encompass an assortment of modalities such as music therapy, herbs, supplements, guided imagery, music therapy, chiropractic, yoga [44], and robot use [45]. Among stressful situations, pain is the most significant clinical issue for which people are forced to turn to complementary interventions. The most common complementary therapies used in children's hospitals include music guided imagery, clinical hypnotherapy, meditation, and pet therapy [44]. According to Misra et al. [44], complementary interventions are helpful to manage pain without significant side effects. In the present review, only Saliba et al. [26] showed a significant impact on SC levels by using a model of CD. Three other studies [24,25,27] failed to demonstrate statistically significant differences in the SC levels before and after intervention.

*Study Limitations*

This systematic review has some limitations: (1) most studies included nonconsecutive patients which could create possible bias, (2) there was heterogeneity in the underlying diseases/interventions, (3) the randomization of the patients where it occurred was not blinded, (4) some studies were performed on a pilot basis, and (5) saliva was collected only from the children and not from their parents. It is important to study something similar in the future since there is no research that has studied both parents and children in the context of a stress reduction intervention in a hospital environment.

## 5. Conclusions

Overall, there is evidence that non-pharmacological interventions hold a promising role in reducing levels of anxiety or distress in children as confirmed with saliva cortisol. However, research on saliva cortisol as a tool of anxiety measurement requires higher quality studies to strengthen the evidence base.

**Author Contributions:** Conceptualization, M.G.; methodology, M.G. and N.Z.; software, A.D. and N.Z.; validation, E.I.K., A.D. and E.M.; investigation, A.A. and E.V.; data curation, A.C.; writing—original draft preparation, M.G., E.I.K., and N.Z.; writing—review and editing, E.I.K., A.D., E.M. and N.Z.; supervision, N.Z.; project administration, N.Z. All authors have read and agreed to the published version of the manuscript.

**Funding:** This research received no external funding.

**Institutional Review Board Statement:** Not applicable.

**Informed Consent Statement:** Not applicable.

**Data Availability Statement:** The data in this article will be shared upon reasonable request to the corresponding author.

**Conflicts of Interest:** The authors declare no conflict of interest.

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
