# Peer review of "Investigating Non-Pharmacological Stress Reduction Interventions in Pediatric Patients Confirmed with Salivary Cortisol Levels: A Systematic Review"

_pediatrrep, doi:10.3390/pediatric15020031_

Round 1

Reviewer 1 Report

In their systematic review "investigating non-pharmacological stress reduction interventions in pediatric patients confirmed with salivary cortsisol levels" Grigoropoulou et al. give an overfiew on the actual literature on non-pharmacological interventions and their impact on children´s anxiety and distress levels.

Overall the manuscript is well written, easy to read and of interest to the reader.

However, some minor points should be adressed:

1) Introduction:  The introduction should include information on the effect of recurrent medical interventions on a childs distress level. Moreover it should include the role of parent behaviour on a childs distress levels. Recurrent traumatic interventions may also lead to impaired parent child interaction an ongoing impairment of child development.

2) Population: please comment on the chosen age range. Is there a reason for excluding children <3y?

3) Figure 1: Shows 4 Databases used. In the text only two Databases are mentioned.

4) 3.3.1 Educational interventions: this paragraph should include information if and how cortisol levels were measured.

5) Page 6, Line 177  - please change "were" to "was"

6) Discussion: it should be discussed if for propper interpretation SC couples of children AND parents would be more informative! Have any of the reviewed studies used SC couples?

Ok

Reviewer 2 Report

Authors review and analyse some non-invasive non-pharmacological interventions in paediatric patients matching them with biochemical method of cortisol detection in saliva. Literature data were analysed from 9 papers after massive exclusion criteria application.

This paper is well written, thus could be accepted with minor revision, if the authors will decide to add some comments in the text or respond comprehensively to some small methodological questions.

Questions:

Results, line 106. Why authors put “…case reports, case series…” in exclusion criteria?

Results, line 106. Why “children needing dental care” is exclusion criteria?

Figure 1, in “exclusion” right part of the flow chart. “Full articles excluded with reasons (N=39) Non meeting the inclusion criteria (n=39). It looks like 39+39=78 papers were excluded. It’s impossible, starting from 53 articles.

Figure 1, in “exclusion” right part of the flow chart. “Comparison of SC… and others”, n=5, n=1, n=3, n=1, totally looks like 10 articles are excluded and accounts don't add up for the remnant papers. Please correct this issue or explain better

Remark:

Some typos are in Figure 1, as “Science Direct (n-22)” must be corrected

Line 285, for the abbreviation STAI-C the definition is missing.

Reviewer 3 Report

This is a systematic review on effect of various non-pharmacological interventions, which was evaluated by measuring salivary cortisol levels. The review is thoroughly performed and well-written. I recommend this paper for publication as the current form.

Following is a small suggestion.

1.     The table is a bit hard to read. I recommend reducing words in the cells, e.g., “Total sample size” can be replaced with “N”.
